# Laser Microtextured Surfaces for Friction Reduction: Does the Pattern Matter?

**DOI:** 10.3390/ma13214915

**Published:** 2020-10-31

**Authors:** Carmine Putignano, Giuliano Parente, Francisco Josè Profito, Caterina Gaudiuso, Antonio Ancona, Giuseppe Carbone

**Affiliations:** 1Department of Mechanics, Mathematics and Management, Politecnico di Bari, 70126 Bari, Italy; giuliano.parente@poliba.it (G.P.); giuseppe.carbone@poliba.it (G.C.); 2Institute for Photonics and Nanotechnologies-IFN-CNR, U.O.S, Physics Department “M. Merlin”, 70126 Bari, Italy; caterina.gaudiuso@uniba.it (C.G.); antonio.ancona@uniba.it (A.A.); 3Department of Mechanical Engineering, Polytechnic School of the University of São Paulo, 01000 São Paulo, Brazil; fprofito@usp.br

**Keywords:** lubrication, laser micro-texturing, friction reduction

## Abstract

Frictional performances of different textures, including axisymmetric and directional patterns, have been tested in the mixed and the hydrodynamic lubrication regimes. Experimental results, corroborated by numerical simulations, show that the leading parameter is the geometrical pattern void ratio since a large number of dimples offers, at low speed, a trap for debris whereas, at high speed, due to the flow expansion in each micro-hole, fosters a fluid pressure drop, the consequent insurgence of micro-cavitation and, ultimately, the reductions of the shear stresses. Furthermore, in this paper, it is shown that, by means of directional textures, equivalent hydrodynamic wedges can be built up, thus establishing different friction performances depending on the flow direction.

## 1. Introduction

Controlling and optimizing friction and wear between surfaces into contact is the ultimate objective pursued in Tribology research in the last decades [1]. Balancing friction, wear and adhesion within a mechanical system is a complex operation, marked by a very large number of variables whose influence, in most cases, is not explicit. To this extent, tuning surface properties is crucial. Thus, any topographic modification applied to the interacting surfaces and, in particular, the introduction of texture patterns, may entail dramatic performance changes in tribological systems. Surface textures can be fabricated through physical or chemical processes, and be randomly or deterministically distributed. Specifically, an increasingly growing interest is dedicated to understanding, experimentally and theoretically, how surface textures affect the tribological properties of lubricated contacts [2,3,4,5]. From a manufacturing point of view, the technological improvements in Laser Surface Texturing (LST), due to the introduction of femtosecond pulsed lasers, have dramatically enlarged the field of structures that can be potentially realized. Indeed, thanks to the short duration of laser pulses, the ablation process can be uncoupled from thermal effects responsible for the qualitative deterioration of the technique. As a consequence, the formation of manufacturing defects and the occurrence of residual stresses have been substantially reduced, thus enabling the creation of extremely precise patterns, without any further post-processing polishing. Furthermore, LST has been proven to be highly flexible, adaptable and ready for rapid reconfiguration. In order to tailor the friction properties to the specific applications, very complex textures can be generated on a large class of materials, including different metals [6,7] and rubber-based composites [8] which often exhibit non-linearly elastic properties [9,10,11]. For these reasons, LST is increasingly becoming a valid alternative to other more widespread processes, such as the ion beam, chemical attack techniques, etc. [12].

Focusing on the tribological performance of textured surfaces, it is noteworthy to point out that the macroscopic effect of conveniently designed microstructures is an improvement in the load carrying capacity, wear resistance and friction properties. Another important effect evaluated, in hydrodynamic regime, is the influence on the coefficient of friction (COF) of the effect of microdimple on the peculiar vibrations of the tribosystem [13]. This effect, at high loads and speeds, is reduced by enabling a marked friction reduction [14,15].

Appropriate microstructuring means that the size of the micro-textures has to be correctly tuned to promote, in the hydrodynamic regime, the occurrence of micro-cavitation without creating vortices in each dimple. Indeed, micro-cavitation involves local vaporization of the fluid and, therefore, an average reduction in viscosity. This results in a local reduction of shear stress and, ultimately, in a smaller global friction coefficient [16]. Interestingly, at low speed and high load conditions, i.e., in the boundary and mixed lubrication regimes, the textures can act as lubricant reservoirs and, at the same time, as traps for wear debris [17,18]. In order to provide a more precise theoretical assessment of the role of micro-textures in lubricated contacts, different model based on analytical [19] or numerical solutions of the Reynolds equation [18,20,21], eventually homogenized on statistical basis, have been developed. However, the optimization process of laser manufactured textures is still mainly carried out on a comparative experimental basis. Galda et al. [22] and Vilhena et al. [23] evaluated the effects of hole-shaped textures and laser parameters, respectively. The effects of dimple size, depth, areal density and roughness have also been widely studied in Ref. [22,24,25,26]. In particular, it has been found that the depth-width ratio of the laser-ablated dimples greatly influences the behavior of tribological textures [27].

In this work, we investigate the role that different texture patterns may have on frictional performances. Several microstructures with distinct geometrical patterns and texture densities, herein designated as geometric void ratio, are benchmarked. For each pattern, a Stribeck curve is built up for the hydrodynamic, mixed and boundary lubrication regimes. The analysis of the experimental outcomes is supported, in the hydrodynamic regime, by numerical simulations aimed at assessing the underlying lubrication mechanisms responsible for the obtained results. With regards to the patterns design, two approaches have been followed: one gives axisymmetric textures, whose design has been inspired by patterns existing in nature; the other one is asymmetric and directional. The variations on the axisymmetric dispositions of the textures are mainly aimed at understanding how the combination of texture spatial distribution and geometric void ratio affects friction. On the other hand, the asymmetric directional patterns are developed to take advantage of anisotropies by creating, depending on the flow direction, equivalent convergent or divergent wedges. The paper is structured as follows. In Section 2, we discuss the manufacturing processes, the tribological measurement setup and we briefly summarize the main aspects of the mathematical modelling and solution methods employed in the numerical simulations. In Section 3, we discuss the experimental outcomes and we corroborate our conclusions with experimental simulations; final remarks are included in the Conclusion section.

## 2. Materials and Methods

### 2.1. Laser Texturing with fs Pulses

The laser equipment employed in this work is a chirped culse cmplificator (CPA), Yb:KGW femtosecond system from Light Conversion (Pharos PH1-SP-1.5 mJ, Vilnius, Lithuania), delivering an almost diffraction-limited beam ($M^{2}<1.3$) at 1030 nm. The pulse FWHM can be changed in the range between 190 fs and 10 ps, the maximum pulse energy is 1.5 mJ and the maximum average power is 6 W. The repetition rate can be varied from a single pulse operation to 1 MHz. The beam exiting the laser system passes through a half wave-plate followed by a polarizer used to adjust the laser power finely. Consequently, the vertical beam polarization is converted into circular by using a quarter wave-plate. Therefore, a system of mirrors directs the laser beam towards a galvo-scanner (IntelliScan 14 from ScanLab GmbH, Puchheim, Germany), equipped with a 56 mm focal length telecentric F-theta lens, which focuses it onto the sample placed on a three-axis micrometre stage. Figure 1 shows the experimental setup used for surface texturing. The laser beam spot size on the sample surface ($2w_{0}$) is approximately 24 µm ($1/e^{2}$).

The laser source is operated at a repetition rate frequency (RR) of 100 kHz to ensure high processing speed, while preventing heat accumulation which would cause melting and hence would be detrimental for the accuracy of the microstructures. The scanning strategy to generate the dimples consists of moving the laser beam along parallel horizontal lines at a scan speed (*v*) of 100 mm/s and hatch distance of 2 µm, for all the fabricated geometries. In such a fabrication process, where the accuracy of the final structures is crucial for the performances of the samples, it is important to choose a working fluence close to the ablation threshold. Therefore, the number of pulses *N* impinging on the same focal area along the scan path has been determined as [28]:(1)$$N=\frac{2w_{0}}{\left(v/RR\right)}$$
to take into account the incubation effect occurring during the multipulse laser irradiation of material, which causes the ablation threshold to significantly decrease as the number of pulses delivered to the sample increases [29]. The ablation threshold has been calculated as:(2)$$F_{th}\left(24\right)=F_{th}\left(1\right)\times N^{S-1}=0.16\;{J/\mathrm{cm}}^{2}$$
by using the Jee model for incubation [30]. $F_{th}\left(1\right)$ is the threshold fluence for a single pulse, *S* is the *incubation coefficient* whose values are in the range between 0 and 1 (S=1 means absence of incubation).

According to the pattern to be fabricated, the applied power *P* has been found between 90 mW and 140 mW, which corresponds to a pulse energy $E_{p}$ ranging from 0.9 to 1.4 µJ. In particular, the Nautilus, the two grids, the Diagonal and the Hydrostep were fabricated at 1.3 µJ, the fishbone at 1.4 µJ and the Sunflower at 0.9 µJ. These discrepancies on the pulse energy used for fabricating the different patterns of dimples characterized by the same depth are ascribable to the sample variability and to the texture dimples slightly dissimilar dimensions, e.g Sunflower is formed by smaller dimples. Therefore, the laser fluence impinging on the sample has been determined as:(3)$$F=\frac{2E_{p}}{\pi w_{0}^{2}}$$
resulting between 0.40 and 0.62 J/cm${}^{2}$, i.e. nearly above the ablation threshold, thus ensuring high reliability of the machining process.

The dimple textures were fabricated on spherical caps produced from 100Cr6 steel spheres to provide a circular flat surface with a diameter of approximately 5 mm. We observe that such a material is a stainless steel, which is chemically inert and easy to work. On the spherical cap base, whose root-mean-square surface roughness after polishing is 30 nm, circular and elliptical dimples have been fabricated depending on the different texture patterns and densities investigated (see Figure 2). The scanning strategy consisted in moving the laser beam along horizontal lines with an hatch of 2 µm and repeating such scan for a number of loops equal to 5, in order to reach the final desired depth. Several patterns, both isotropic and anisotropic, were considered. The designs presented several differences, both in terms of dimples arrangement on the surface and geometric void ratio, as summarized in Figure 3. Incidentally, let us observe that the geometric void ratio (VR) is here defined as the ratio between the surface occupied by the dimples and is determined as the ratio between the total machined surface area (sum of each circle and/or ellipse area, depending on the geometry of the dimples) and the total circular area of the truncated sphere. The dimples’ depth was set constant and equal to 6.5 µm.

#### 2.1.1. Axisymmetric Textures

**Nautilus Texture**—The first texture has been inspired by the nautilus and it includes circular dimples with a diameter of 180 µm. The Cartesian coordinates of the dimples’ centres were selected by using the parametric equations:(4)$$x\left(\theta \right)=a\times e^{b\theta }\cos\theta$$(5)$$y\left(\theta \right)=a\times e^{b\theta }\sin\theta$$
where a=250 µm, b=0.08 is the cotangent of the angle between the radial line and the tangent line in each considered point of the spiral. θ identifies the angular position of each point of the spiral (starting from 0). The angular step for going from one dimple to the adjacent one was $\Delta \theta =0.3^{\circ }$. The geometric void ratio of the Nautilus texture was 13% (see Figure 3a).**Sunflower Texture**—Further inspiration has been taken from nature to replicate the sunflower inflorescence. In this case, the pattern includes circular and elliptical dimples whose dimensions are listed in Table 1. The dimples are arranged along 60 diametric directions. Consequently, the angular distance between adjacent diametric lines of dimples is $6^{\circ }$. Moreover, there has been a spatial shift of 75 µm between two adjacent lines. The positions of the n-th dimple along the diametric lines have been chosen according to the equation:
(6)$$y_{n}=y_{0}+\Delta_{0}+n\times d$$
where $y_{0}=600$ µm is the distance from the centre of the sample of the first dimple, i.e. the radius of the untextured surface, $\Delta_{0}=150$ µm and d=0.82 µm. For the first dimple, it was chosen:(7)$$y_{1}=y_{0}+\Delta_{0}$$

The geometric void ratio of the Sunflower texture was 20% (see Figure 3b).**Uniform Grid Textures**—The last two patterns have been characterized by uniform homogeneous matrix of 180 µm circular dimples. In order to have different geometric void ratios (33% and 44%), the mutual distance between the dimples has been fixed equal to 278 µm and 239 µm, respectively.

#### 2.1.2. Directional Textures

**Diagonal Texture**—In the Diagonal pattern, the distance between circular dimples having a diameter of 180 µm increases in both *x* and *y* directions, according to the following equations which identify the positions of the n-th dimple along the *x* and *y* directions:(8)$$x_{n}=n\times x_{1}+\frac{a}{2}\left(n^{2}-n\right)$$(9)$$y_{n}=n\times y_{1}+\frac{b}{2}\left(n^{2}-n\right)$$
where a=b=25 µm and $x_{1}=y_{1}=250$ µm. The overall effect has been a gradual distancing in the diagonal direction between *x* and *y* axes. The origin of the coordinate system has been placed at top right corner, outside the circular region of the sample. The geometric void ratio of the Diagonal texture was 14% (see Figure 3e). **Hydrostep Texture**—The Hydrostep geometry has been conceived similarly to the diagonal one. However, in this case, the origin of the coordinate system is at the top centre. Therefore, in the *x*-direction, there has been a symmetric increasing distance between dimples along both sides. At the same time in the *y*-direction, there is no symmetry and the distance continuously increases from the top downwards. The positions of 180 µm diameter dimples have been identified by Equations (Equation 8) and (Equation 9). The geometric void ratio of the Hydrostep texture was 20% (see Figure 3f).**Fishbone Texture**—The last texture, i.e. the Fishbone, has been constituted by a central part having lines of 180 µm diameter dimples with an increasing distance along the *x*-direction (according to Equation (Equation 8)), symmetrically arranged along both sides. Instead, the distance has been kept fixed and equal to 250 µm in *y*-direction. The side parts include elliptical dimples, having a minor axis of 128 µm and a major axis of 266 µm. Moreover, such ellipses have been rotated by an angle of $\pm 45^{\circ }$ with respect to the *x*-direction. The geometric void ratio of the Fishbone texture was 30% (see Figure 3g).

### 2.2. Morphological Characterization of the Laser Textured Patterns

The morphology of the textures has been evaluated through optical and confocal microscopy.

In particular, in Figure 3, we show the optical microscope images of the textures fabricated on the spherical caps. It can be noticed that the fabricated patterns correctly follow each designed pattern. Moreover, the quality of the microstructures obtained by laser ablation is highlighted in Figure 4, where the top and bottom of a representative circular dimple are shown. Here, it is quite clear that the dimple edges do not present any evident burr and its dimensions deviate from the design less than 1%. The bottom of the dimple presents a regular roughness, without any apparent defect.

A further investigation of the laser fabricated textures and the dimples final depth has been carried out by performing confocal microscope images. The confocal microscope used is a CSM-Instruments (Peseux, Switzerland) with a lateral and vertical resolution of, respectively, 1.1 µm and 0.005 µm. For this purpose, the 3D rendering of a circular dimple has been performed, as shown in Figure 5a,b.

Here, it can be seen that the dimple does not have a perfect cylindrical geometry but rather a slightly conical shape, with some deeper regions at the bottom edges. In order to account for such small deviations from the design and to calculate the average depth of the microstructures, an algorithm determining the height level of the dimple with respect to the untreated surface has been developed in Python. Then, a histogram presenting two peaks centered at different heights, i.e. the depth of the investigated dimple and the reference level of the untreated surface of the sample, has been built, as shown in Figure 5c. Here, the more the dimple depth differs from the average, the broader the corresponding peak in the histogram is. Therefore, the histogram peaks width provides an indication of the deviations from the ideal cylindrical shape of the dimples. This could be illustrated in the histograms (see suggestions in Figure 5c and Figure 6b. Further statistical analysis of the textures features, especially the microstructures depth, has been carried out by taking confocal microscope images of a wider area, thus catching several dimples, as shown in Figure 6.

In Figure 6a it is visible that, as also observed in Figure 4, the textured regions are completely melting and burr-free. Also in this case, a histogram has been built, collecting the height levels of the dimples and the reference surface, as depicted in Figure 6b. The average depth, determined as the difference between the height levels of the two peaks, is equal to 6.5 µm, as for the single dimple, thus further confirming the high reliability of such fabrication process.

### 2.3. Tribological Characterization

The tribological characterization of the LST samples has been carried out with a pin-on-disc tribometer (mod. THT, CSM Instruments, Peseux, Switzerland). The equipment consists of a rotating disc (aluminium alloy AA6061
T6 with a surface roughness of Sa =80 nm) that has been kept into contact against the laser textured spherical caps. The samples have been assembled in such a way that the contacting surfaces were aligned in parallel to each other. The contact region has been immersed in a lubricant bath, whose temperature has been monitored continuously during the test. The lubricant used in the experiments was a mineral base oil (Oroil Therm7 by Orlando Lubrificanti S.r.l., Argenta, Italy), whose temperature dependence of dynamic viscosity has been determined using the following polynomial expression:(10)$$\ln\left[\frac{\eta (T)}{\eta_{0}}\right]=\sum_{k=1}^{3}a_{k}{\left(T-T_{0}\right)}^{k}$$
with $\eta_{0}=1.51$ Pa·s, $T_{0}=0.01^{\circ }$ C, $a_{1}=-7.58\times 10^{-2}$, $a_{2}=2.82\times 10^{-4}$ and $a_{3}=-3.76\times 10^{-7}$.

The coefficient of friction (COF) between the surfaces has been measured according to the following test protocol. For each sample, the friction measurements have been conducted in two steps, each consisting in measuring the COF as the sliding speed varies, setting the lubricant temperature and the load. The latter has always been taken equal to 1 N. The measurements have been carried out for 12 different sliding speeds (*v*) ranging from 0.502 m/s to 0.006 m/s, starting from the highest speed. This practice has been adopted to strongly minimize any possible surface wear as, for the highest speed values, full-film hydrodynamic regime occurs and, consequently, the two surfaces are not in contact at all. The first test step has taken place at room temperature (this one has always been recorded to assess the reference viscosity value). The second test step has been carried out by sweeping the same sliding speed range at the temperature 120 ${}^{\circ }$C to evaluate friction for smaller viscosity values. This protocol ensures the possibility of characterizing the entire Stribeck curve.

To better investigate the possible effects of asymmetries, it has been decided to take each measurement by fixing the speed and rotating the disc clockwise and counterclockwise. In the absence of significant asymmetries, the two obtained COF values were averaged. In the case of asymmetric textures, they have been assessed separately. All measurements have been performed by maintaining stable friction conditions for no less than 20 s.

Each COF value plotted on each sample’s Stribeck curve has been obtained by averaging four consecutive measurements. The measured value has been processed by eliminating the contributions of any instrument offsets or viscous friction only due to the motion of the sphere through the lubricating fluid. For each data set, new samples and new lubricant have been employed. All parts of the apparatus in direct contact with the lubricant have always been washed with abundant distilled water, followed by alternate use of isopropanol and acetone. The moving parts have also been washed in an isopropanol ultrasonic bath.

### 2.4. Finite Volume Numerical Simulation

The hydrodynamic problem governing the lubrication behaviour of the textured surfaces studied in this paper has been tackled by using the numerical simulation framework proposed by Profito et al. [31] for general journal bearing systems. Such a framework has been already successfully employed to evaluate the lubrication performance of sliding textured contacts [32,33]. In the following, only pertinent aspects of the mentioned simulation framework applied to the pin-on-disc contact configuration considered in this work are described.

As depicted in Figure 7, the lubricated conjunction under investigation can be assumed as a conformal parallel sliding bearing operating under full-film hydrodynamic lubrication regime. For these conditions, the steady-state isothermal Reynolds equation with the Elrod-Adams p−θ mass-conserving cavitation model can be written as
(11)$$\begin{array}{c} \frac{\partial }{\partial x}\left(\frac{\rho h^{3}}{12\mu }\frac{\partial p}{\partial x}\right)+\frac{\partial }{\partial z}\left(\frac{\rho h^{3}}{12\mu }\frac{\partial p}{\partial z}\right)=\frac{\partial }{\partial x}\left(\frac{\rho \theta Uh}{2}\right), \end{array}$$
with the complementary boundary conditions for cavitation
(12)$$\begin{array}{c} \left(p-p_{cav}\right)\left(1-\theta \right)=0\to \left\{\begin{array}{cc} p>p_{cav},\;\theta =1 & \mathrm{in}\;D^{+}\\ p=p_{cav},\;0\leq \theta <1 & \mathrm{in}\;D^{-}\\ p=p_{cav}\; & \mathrm{on}\;C. \end{array}\right. \end{array}$$

In the above equations, p(x,z) is the fluid pressure, θ(x,z) the lubricant film fraction (cavitation), $p_{cav}$ the cavitation pressure, h(x,z) the oil film thickness, *U* the sliding velocity, and μ and ρ the lubricant dynamic viscosity and density, respectively. Both viscosity and density were assumed constant in the analysis. Moreover, $D^{+}$ and $D^{-}$ are the corresponding pressured and cavitated regions, and C the cavitation boundaries.

By assuming no misalignment between the surfaces, the lubricant film thickness can be expressed in the coordinate system Oxyz attached to the stationary pin (see Figure 7) as
(13)$$\begin{array}{c} h\left(x,z\right)=h_{0}+h_{T}\left(x,z\right), \end{array}$$
where $h_{0}$ is the minimum oil film thickness provided by the rigid body displacement between the surfaces in the y-direction, and $h_{T}\left(x,z\right)$ is the local texture’s depth. Notice that, given the low applied load (1 N) of the tests, it was assumed that solid deformation is negligible.

In order to reproduce the steady loading conditions imposed in the pin-on-disc tests, the following equilibrium equation should be satisfied in the y-direction
(14)$$\begin{array}{c} W_{H}\left(h_{0}\right)=F_{ext}\;\to \;\int_{-L_{z}/2}^{L_{z}/2}\int_{-L_{x}/2}^{L_{x}/2}p(x,z)dxdz \end{array}$$
where $W_{H}$ is the hydrodynamic load-carrying force, $F_{ext}$ the externally applied load, and $L_{x}$ and $L_{z}$ the contact length and width, respectively. Similarly, the corresponding hydrodynamic friction force acting on the pin surface, and the associated coefficient of friction (COF) are calculated as
(15)$$\begin{array}{cc} F_{H} & =\int_{-L_{z}/2}^{L_{z}/2}\int_{-L_{x}/2}^{L_{x}/2}\left(\frac{h}{2}\frac{\partial p}{\partial x}-\frac{\mu \theta U}{h}\right)dxdz, \end{array}$$
(16)$$\begin{array}{cc} COF & =\frac{F_{H}}{W_{H}}. \end{array}$$

The numerical solution of Equation (Equation 11) was implemented using the hybrid-type Element-Based Finite-Volume Method (EbFVM) proposed by Profito et al [34], which allows the solution of lubrication problems in the presence of cavitation on unstructured meshes. The principal benefit of the EbFVM is to provide an integrated, standardized and consistent methodology that combines the geometric flexibility of the Finite Element Method (FEM) to deal with complex geometries and the inherent conservative nature of the Finite Volume Method (FVM). The latter aspect is essential for solving lubrication problems with cavitation properly since it automatically guarantees the local and global flow conservation directly in the discrete formulation. Hence, for the Elrod-Adams p−θ cavitation model in consideration, the complementary Jakobson-Floberg-Olsson (JFO) boundary conditions [35,36] and the associated fluid transport conservation throughout the lubricated domain, particularly on the cavitation boundaries, can be more readily treated with the EbFVM. Therefore, while the FEM could also be used to solve lubrication problems with complex geometries and irregular grids, when fluid-film cavitation is crucial, the EbFVM provides a more straightforward discretization scheme and computational implementation advantages.

The determination of the equilibrium minimum oil film thickness ($h_{0}$) which balances a given applied external load ($F_{ext}$) was accomplished by solving the nonlinear equilibrium equation in the y-direction (Equation (Equation 14)) using the Newton-Raphson method with Armijo’s line search technique. The reader is referred to Refs. [34,37] for more details about the mathematical modeling and solution framework, including experimental validations and applications to transient problems involving textured surfaces.

## 3. Results and Discussion

In Figure 8, for the non-directional textured patterns, we plot the coefficient of friction (COF) as a function of Hersey number $H=\eta v/P_{av}$, with $P_{av}$ being the average contact pressure defined as $P_{av}=F/\pi a^{2}$, with *F* and *a* being, respectively, the load and the of the spherical cap circular basis radius. Specifically, we compare the frictional performances of the untextured sample with the Nautilus, the Sunflower, the Fishbone and the two homogeneous (33% and 44% geometric void ratios) textures.

In the mixed lubrication regime ($10^{-9}<H<10^{-7.5}$), we notice that only the textures with large geometric void ratios (Fishbone, 33% Grid and 44% Grid) provided significant advantages in terms of friction. Indeed, there occur two competing mechanisms: on the one hand, the pattern of dimples increases the surface irregularities, thus potentially worsening the lubrication effects and hence the frictional performances; on the other hand, dimples can act as debris traps, lubricant reservoirs and local reducers of resistive shear forces, thus contributing to improving the overall friction.

The latter two beneficial effects seem to prevail when the geometric void ratio grows, thus suggesting, at low speed, the effectiveness of structures with high-density textures.

In the full-film hydrodynamic regime ($10^{-7.5}<H<10^{-5.5}$), we have a different scenario as all the textures show a marked improvement in terms of frictional performances. In particular, friction decreases as the geometric void ratio increases. Such a trend is coherent with results in the literature [38] and, as we will later explain in detail, can be related to the occurrence of micro-cavitation [39]. Let us preliminarly observe that, in agreement with the results shown in Ref. [30] for a different shape of dimples, in the velocity range investigated, the performance enhancement tends to reach a saturation value for large geometric void ratios. Indeed, the Fishbone pattern and the two uniform meshes provide similar friction coefficients.

In order to physically explain these results and, specifically, to relate the presence of dimples to the occurrence of micro-cavitation and, thus, to the friction reduction, we have employed the numerical modelling based on the finite volume solution of the Reynolds equation with a mass-conserving cavitation model previously described. By means of such a technique, the Reynolds equation is solved, in the hydrodynamic regime, accounting for the presence of a biphasic flow in the regions where cavitation occurs. Let us notice that, given the low applied load, we assume that solid deformation is negligible. To evaluate the reliability of the numerical modelling for the texture patterns under investigation in this work, in Figure 9, we compare, with a relatively good agreement, the experimentally measured COF with the ones obtained from simulations.

Now, the numerical methodology can be employed to assess what happens locally in the contact region. In particular, let us focus our attention to the film pressure, to the fluid fraction, defined as $\theta =\rho_{bf}/\rho_{lub}$, i.e., the ratio between the biphasic mixture density and the lubricant density, and, ultimately, to the average shear stress. For the uniform texture with 44% as geometric void ratio and a Hersey parameter equal to $H=2.35\times 10^{-6}$, the contour plots of these quantities are presented in Figure 10a.

From a macroscopic point of view, as we are dealing with a uniform mesh realized on a sphere with a flattened region, we tend to have larger values of pressure in a central band parallel to the flow speed; accordingly, here, the fluid fraction and the shear stresses are both larger. The really interesting effects are, however, local and occur within and in the vicinity of each dimple. Indeed, each micro-cavity promotes a local increase in the film thickness, as can be seen in Figure 11. In this case, the lubricant tends to cavitate at the dimple’s inlet (divergent zone) with a drop in both fluid pressure and fluid fraction (film rupture). Subsequently, as the fluid approaches the dimple’s outlet (convergent zone), as expected, the fluid pressure increases and the lubricating film is reformed. With regards to the average shear stress, it experienced a significant reduction over the entire dimple due to the local expansion of the film thickness (lower shear rates) combined with the smaller values of film fraction at the dimple’s inlet (reduced effective viscosity).

This local effect is strictly related to the micro-hole and does not depend on the particular global pattern of the texture. This is confirmed in Figure 10b, where we show, for the Sunflower texture, the contour plots of the same flow quantities. Also, in this case, there exist macroscopic effects related to the global texture pattern, i.e. the pressure increases in the central portion of the contact, where the fluid fraction is larger and the average shear stress reaches its maximum value. Locally, the dimples still carry out their beneficial outcomes as they induce a local increase of the film thickness and the occurrence of micro-cavitation.

We may conclude at this point that, fixed the dimple dimensions, the frictional performance of axisymmetric non-directional texture patterns depend mainly on the texture geometric void ratio since the friction reduction is local and related to the presence of each micro-hole. Conversely, when dealing with non-symmetrical textures, friction outcomes can significantly differ. Let us start by rotating the Fishbone texture, as shown in the inset in Figure 12. In this case, the flow runs initially over the elliptical rotated micro-holes. As shown in the literature [25], this leads to a deterioration of the friction performance as the occurrence of micro-cavitation is affected.

Even more interesting is the case of directional textures. Indeed, as depicted in Figure 13a, a directional texture can be associated with an equivalent wedge. Thus, depending on the flow direction, the pattern behaves like an equivalent convergent or divergent structure.

In this case, unlikely what we have previously seen for the axisymmetric textures, the pattern and the flow direction do play a significant role as, in addition to the local influence of each dimple, macroscopic effects are significant. This is indeed true in the hydrodynamic regime where, an equivalent convergent wedge tends to offer larger hydrodynamic support, with higher equilibrium minimum oil film thickness and lower friction value. In contrast, a divergent profile leads to a reduction of hydrodynamic support and higher friction. This is experimentally confirmed in Figure 13b, where, for the Diagonal texture, we invert the flow direction, thus exploring both the convergent and the divergent wedge cases. As expected, the latter has larger friction. At very large speed values, the two curves tend to collapse as the local effects, related to micro-cavitation and local shear stress over the dimple, tend to prevail. Similar behavior is found in Figure 14, where the Hydrostep texture is tested in the equivalent convergent and divergent wedge configuration. Also, in these conditions, the equivalent divergent wedge shows significant deterioration of frictional performances. Interestingly, we have rotated the pattern obtaining a more symmetrical configuration, whose friction values are intermediate between the convergent and divergent cases.

Ultimately, when employing directional textures, the beneficial effects, related to the micro-cavitation and local film thickness increase due to the presence of each micro-hole, still occur. Nevertheless, it is worth mentioning that the global pattern may determine different frictional performances depending on the flow direction. This may have crucial implications in all the engineering applications where, in presence of relative sliding motion, it is required to change the friction properties when changing the sliding direction.

## 4. Conclusions

In this paper, we have investigated the role that the pattern geometry has in the definition of the frictional performances of laser micro-textured structures. Specifically, we have tested a variety of structures, being symmetrical or marked by a directionality, but all with comparable dimensions of the dimples. From a manufacturing point of view, laser texturing has proved to be a versatile and effective technique allowing high accuracy in the fabrication of micro-dimples. Given the technical capabilities of the manufacturing process, one can then wonder if there exists any optimal pattern for the dimples and, more generally, what is the design criterion to follow when designing a texture.

To this aim, we have initially focused on symmetric textures. These include the Nautilus, the Sunflower and the Fishbone patterns and two uniform meshes with different geometric void ratios. Indeed, in the mixed lubrication regime, we have shown that textures with high geometric void ratios improve the tribological performances as the micro-holes offer space for wear debris trapping and behave as lubricant reservoirs and local reducers of resistive shear forces. Furthermore, at larger speeds, i.e., in the full-film hydrodynamic lubrication regime, the geometric void ratio has still revealed to be the crucial parameter governing the tribological performances as the global friction reduction is due to the local presence of each dimple. Indeed, as shown by numerical analyses, in each micro-dimple, the local increase of film thickness, assisted by the occurrence of micro-cavitation, determine a reduction of the average shear stress. This means that, for symmetric textures, the pattern geometry plays a residual role and the geometric void ratio should be maximized for a friction reduction, which, in comparison with the untextured sample, may go up to 60%. Therefore, it is noteworthy that, coherently with other studies in the literature [30], the optimization process of symmetrically distributed textures should be mainly focused on determining the most effective texture sizes that promote the best tribological performances.

Different behavior occurs when dealing with directional textures. In these cases, we still have a local reduction of the average shear stress over each dimple. However, at a macroscopic level, the directional pattern behaves like an equivalent convergent or divergent wedge depending on the flow direction. Particularly, when the texture pattern acts as a convergent wedge, the hydrodynamic support is enhanced and, thus, friction is reduced. Conversely, in the divergent case, the hydrodynamic support is lower, and friction is higher. Such a directional functioning can be, obviously, adequately tuned to have technological relevance in applications requiring distinct frictional properties according to the sliding directions of the lubricated surfaces.

## Figures and Tables

**Figure 1 materials-13-04915-f001:**
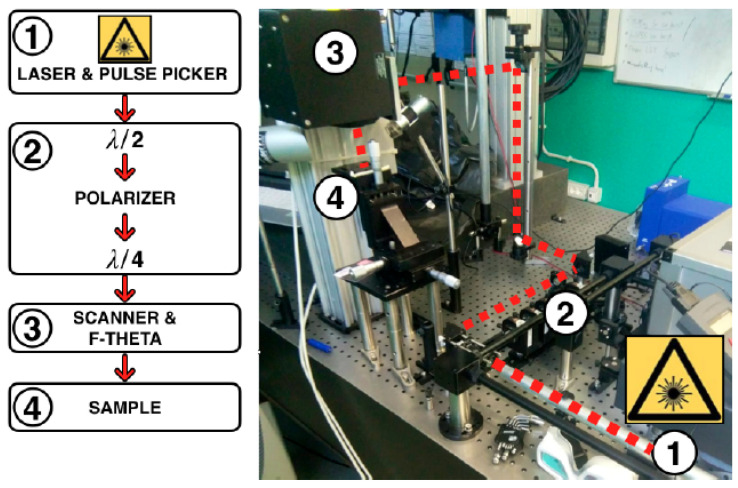
Experimental setup: from the pulse picker (1), the laser beam passes through a half wave-plate followed by a polarizer used to adjust the laser power finely, then its vertical polarization is converted in circular by using a quarter wave-plate (2). A system of mirrors directs the laser beam towards a galvo-scanner (3) which focuses the beam onto the sample (4).

**Figure 2 materials-13-04915-f002:**
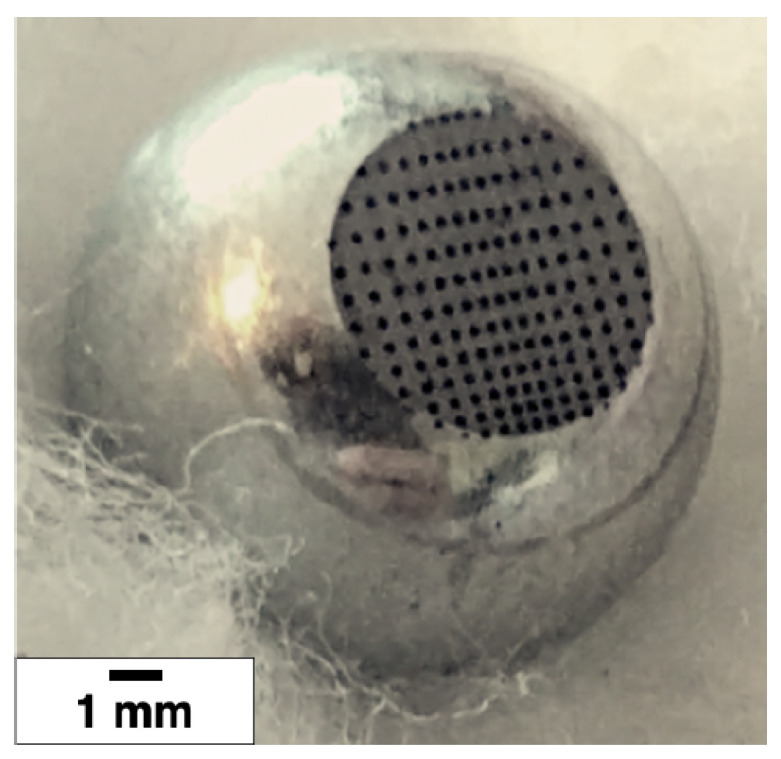
Microtextured sample-Diagonal texture.

**Figure 3 materials-13-04915-f003:**
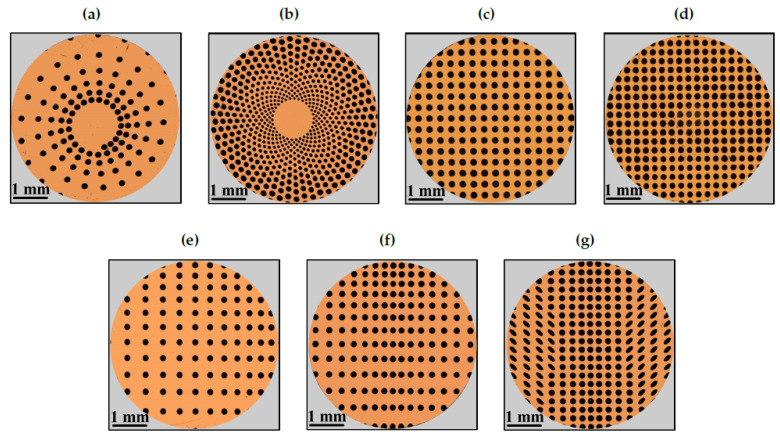
Optical microscope images of patterns fabricated on 100Cr6 steel spheres-(**a**) Nautilus; (**b**) Sunflower; (**c**) Grid (VR=33%); (**d**) Grid (44%); (**e**) Diagonal; (**f**) Hydrostep; (**g**) Fishbone.

**Figure 4 materials-13-04915-f004:**
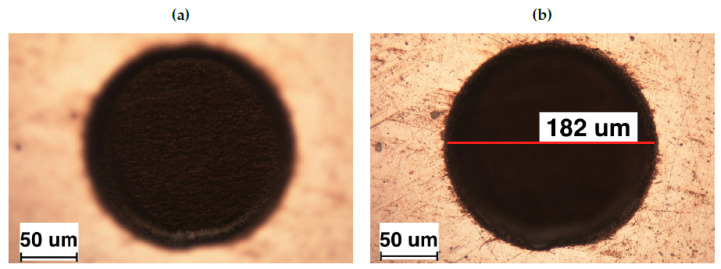
Optical microscope image of (**a**) the bottom and (**b**) the top of a representative dimple.

**Figure 5 materials-13-04915-f005:**
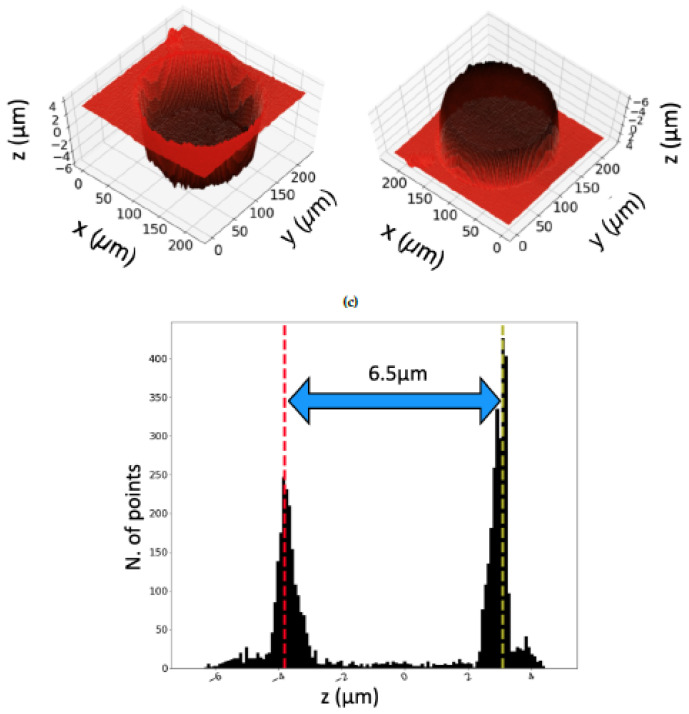
3D rendering of (**a**) the dimple and (**b**) its bottom. In (**c**), the histogram presenting the distribution of the height levels in (**a**) is shown. The distance between two peaks, being equal to 6.5 µm, represents the depth of each dimple.

**Figure 6 materials-13-04915-f006:**
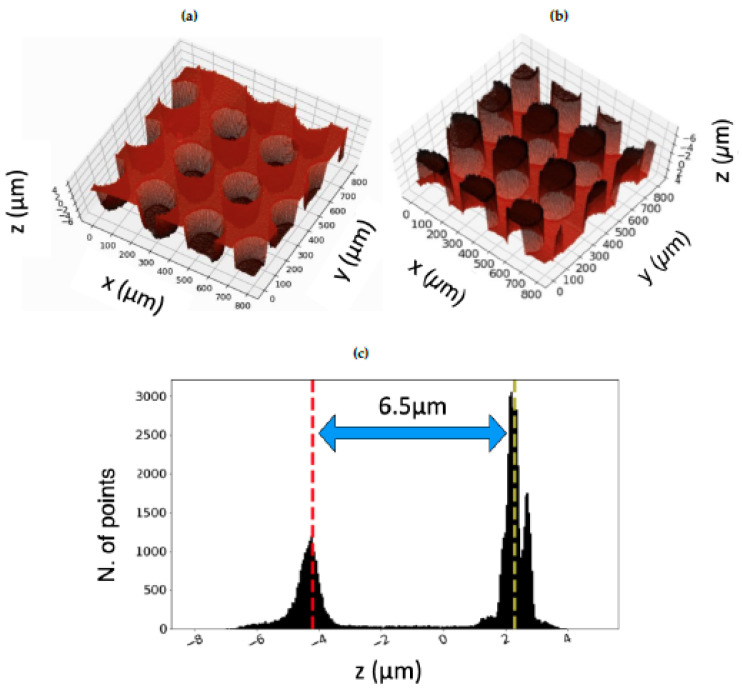
3D rendering of (**a**) Fishbone texture and (**b**) its bottom. In (**c**), the histogram presenting the distribution of the height levels in (**a**) is shown. The distance between two peaks, being equal to 6.5 µm, represents the depth of each dimple.

**Figure 7 materials-13-04915-f007:**
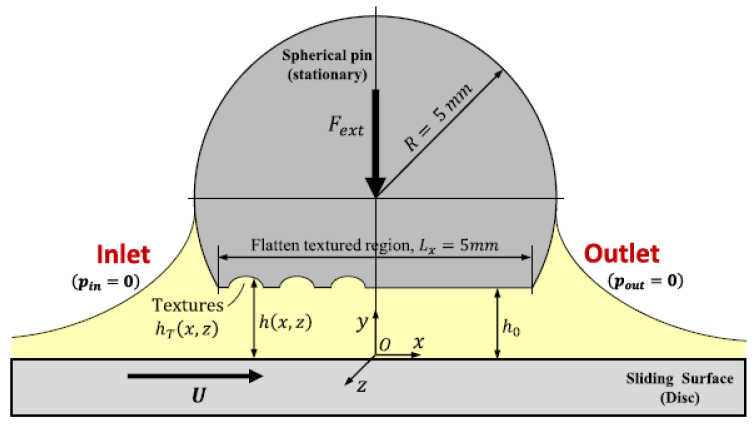
Schematic of the pin-on-disc experimental setup illustrating the textured interface and the main variables used in the numerical modelling.

**Figure 8 materials-13-04915-f008:**
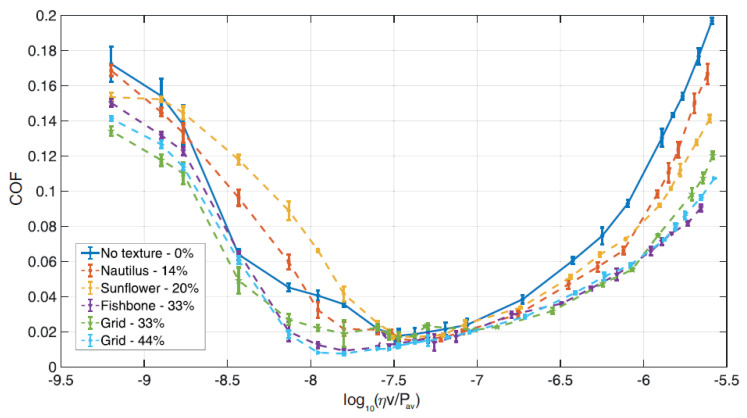
Coefficient of friction as a function of the Hersey number $\log_{10}\left(\eta v/P_{av}\right)$ for the untextured sample, the Nautilus, the Sunflower, the Fishbone and the two uniform (33% and 44% as void ratios) textures. The load is constant and equal to 1 N.

**Figure 9 materials-13-04915-f009:**
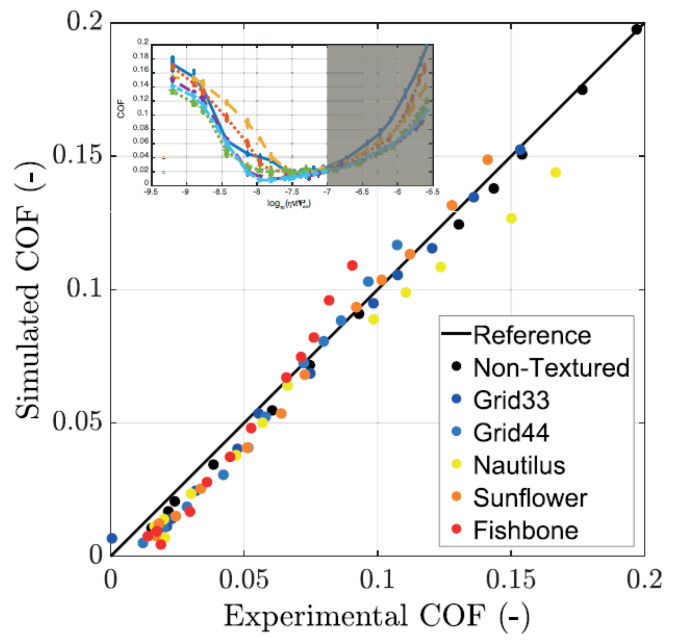
Simulated vs experimentally measured friction coefficient. In the inset, where the experimental Stribeck curve is shown, the speed range, where the comparison is carried out, is shaded.

**Figure 10 materials-13-04915-f010:**
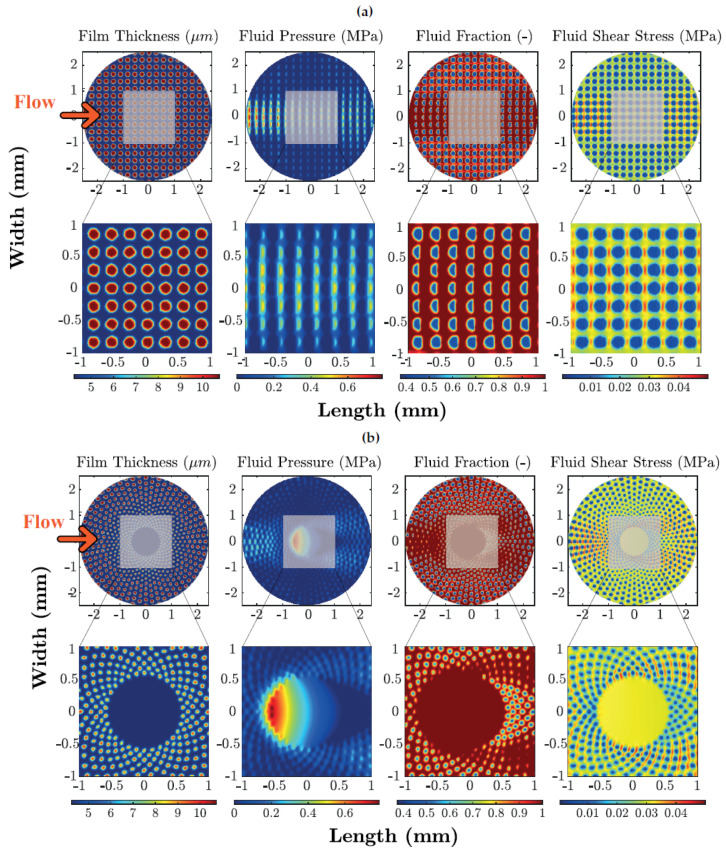
Contour plots for film thickness, fluid pressure, fluid fraction and average shear stress. Results, obtained by means of FV numerical simulations, refer to a Hersey number equal to $\eta U/P_{av}=2.35\times 10^{-6}$—(**a**) Uniform texture (44%), (**b**) Sunflower (20%).

**Figure 11 materials-13-04915-f011:**
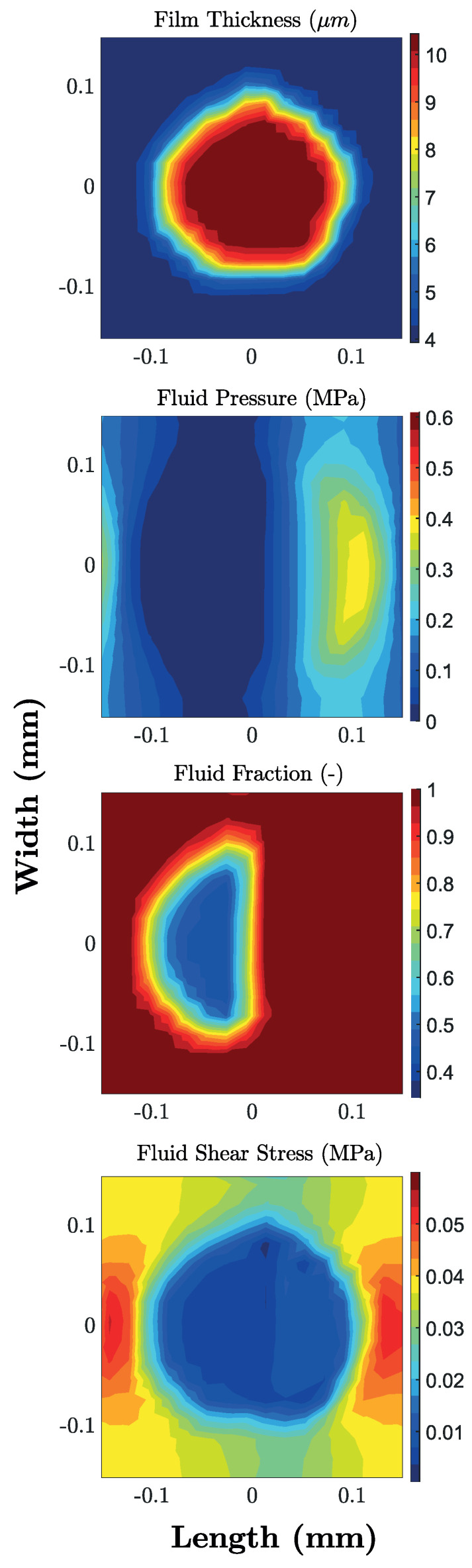
Contour plots for a single dimple, obtained from the uniform (44%) texture, for film thickness, fluid pressure, fluid fraction and average shear stress. Results, obtained by means of FV numerical simulations, refer to a Hersey number equal to $\eta U/P_{av}=2.35\times 10^{-6}$.

**Figure 12 materials-13-04915-f012:**
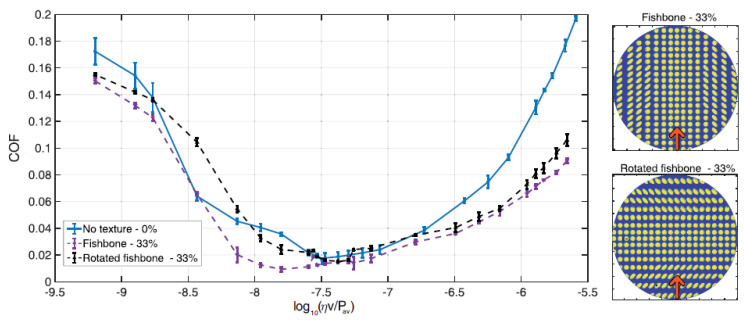
Coefficient of friction (COF) as a function of the Hersey number $\log_{10}\left(\eta v/P_{av}\right)$ for the unxtextured sample, the Fishbone and the rotated Fishbone textures (as shown in the schematics on the right). The load is constant and equal to 1 N.

**Figure 13 materials-13-04915-f013:**
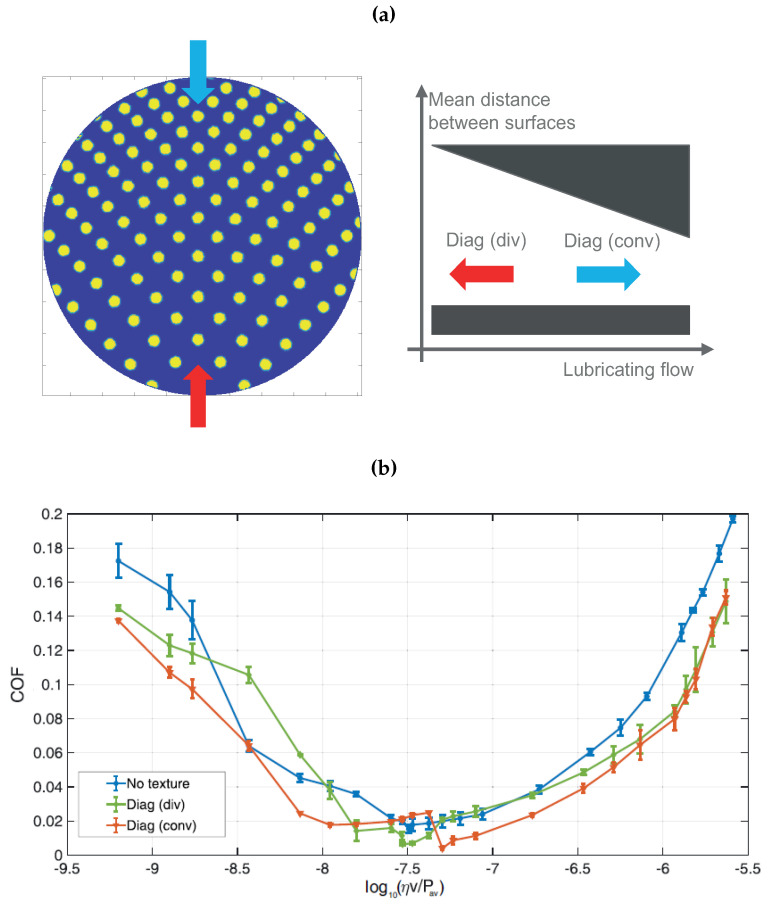
(**a**) Schematic of the Diagonal texture (left) and equivalent convergent and divergent wedge (right), (**b**) Coefficient of friction (COF) as a function of the Hersey number $\log_{10}\left(\eta v/P_{av}\right)$ for the unxtextured sample and the Diagonal texture in convergent and divergent configurations. The load is constant and equal to 1 N.

**Figure 14 materials-13-04915-f014:**
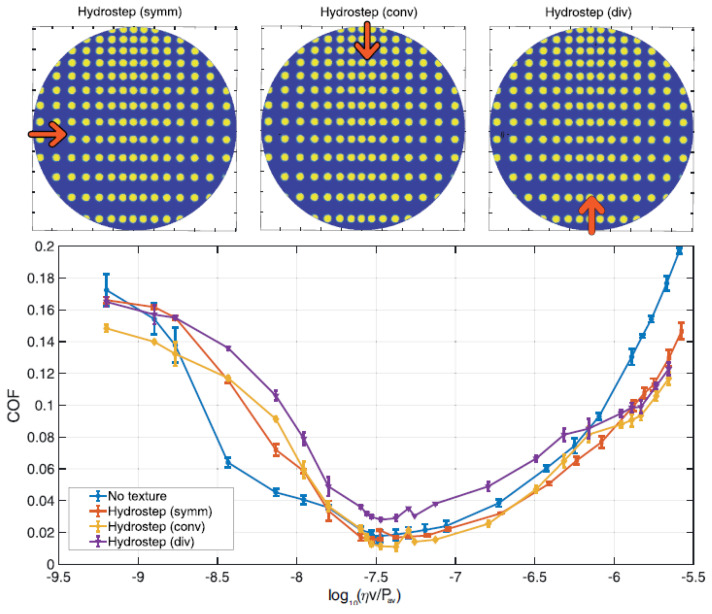
Coefficient of friction (COF) as a function of the Hersey number $\log_{10}\left(\eta v/P_{av}\right)$ for the unxtextured sample and the Hydrostep texture in the symmetrical, the convergent and the divergent configurations. The load is constant and equal to 1 N.

**Table 1 materials-13-04915-t001:** Sunflower’s dimples dimensions.

n	2a (µm)	2b (µm)
1	55	55
2	60	56
3	65	57
4	70	58
5	75	59
6	80	60
7	90	110
8	92	110
9	114	110
10	136	120
11	158	130
12	180	140

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
