# Peer review of "Laser Microtextured Surfaces for Friction Reduction: Does the Pattern Matter?"

_materials, 2020, doi:10.3390/ma13214915_

Round 1

Reviewer 1 Report

Comments for Materials-949204 V1 This manuscript has investigated the frictional performances of different textures, including axisymmetric and directional patterns in the mixed and the hydrodynamic lubrication regimes. In addition, experimental results, corroborated by numerical simulations were represented and discussed. However, there are some questions should be clarified and responded before the consideration of publication. (1) The abstract should be revised to a normal presentation way. For example, ‘we can create equivalent hydrodynamic wedges, thus establishing different friction performances according to the flow direction.’ The word ‘we can’, should be deleted. (2) In this work, the tribological characterization of the samples was carried out with a pin-on-disc tribometer (mod. THT, CSM Instruments, Peseux, Switzerland). However, the real-time COF curves were missed. It is advised to add some typical curves.

Reviewer 2 Report

  1. Section 2.1
    1. Please give the definition of the laser pulse duration und the laser spot size (FWHM, 1/e, 1/e²)
    2. Units and prefixes should not be italic
    3. The authors mentioned a used fluence regime but did not mentioned the actual applied fluence for the individual LST
    4. Please describe the process of LST in detail: number of scans, rotating or constant hatch angle
    5. Line 81: Please clarify the unit of a
    6. Line 91: the void ratio of the Sunflower texture is given as 20%. An objective view on figure 3b implies a higher value of the VR. Please clarify this.
    7. Line 100: The origin of the coordinate system of the Diagonal Texture was given as "bottom left corner" but figure 3e implies that it is on "top right".
    8. Figure 3: Please add the texture description to the subcaption or the caption
  2. Section 2.2
    1. Figure 5: Please provide a cross section of the dimple. We think that the slope of the dimple is crucial for the tribolocical behavior of the texture
  3. Section 2.3
    1. The surface roughness of the rotating disc is given with Ra. Please clarify: Is it the Line roughness Ra or the surface roughness Sa. In Addition: In line 74 the authors use the RMS roughness, please unify the measurement values of the roughness (The reviewer suggest Sa)
  4. Section 3
    1. Due to the LST the contact area (used for calc of H) should be related to the void ratio. Is the reduction of the contact area considerd in the results?
    2. Does the values in the COF(H) charts represent the mean and the diviation of the 20 second test duration or the mean and deviation of the 4-fold test run?
    3. The authors mentioned, that the temperatur of the lubricant is measuerd during the tribolocal evaluation. Does the temperatur differs for different sliding speeds? Is this effect considerd in the calculation of H?
    4. The authors mentioned, that the frictional behavior is tested at RT and under 120°C. Are both measurement conditions are combined in Fig 7,12,13,14? Is the temperatur increase of approx 100°C just responsible for a change in the viscosity or has it also other effects on the tribological behavior?
    5. The titel of the paper arise the question "Does the pattern matter?" For symmetric textures: What is more important?: The pattern or the Void ratio? the authors start the discussion on this question in in line 222 but the experimental data would significantly support this thesis. The reviewer suggest to compare more LST with identical VR to verify your claim.
    6. Please combine Fig 9 and 10 for space saving and higher clearness
    7. Please combine Fig 13 and 15 for space saving and higher clearness
    8. In line 234 the authors mentioned a significant effect of the flow direction: Please reveal the significant effect in the related figure and the relevant H-range. Please explain the underlying process of the effect mor in detail and identify potential application scenarios.
  5. Conclusion
    1. The Authors called their structures "bio-inspired". From the point of the reviewer this term implies, that LST imitates nature-provided structures to solve surface related problems. In this work, the structures just look like something out of the nature (Flower, Nautilus) and thus should not called "bio inspired"

Reviewer 3 Report

The article presents an interesting study of the influence that different texturing patterns have on the friction coefficient. For this purpose, micro cavities of 180 microns in diameter and 6.5 microns in depth are textured with a femtosecond pulsed laser. This is an interesting study that reaches different conclusions, among which the importance of the density of micro-cavities over the geometric arrangement of them stands out. Although it is an interesting article, there are certain points that need should be reviewed:

1-. As is logical, for the study to be approachable, there are certain variables that have been fixed, such as the material (steel 100Cr6), the contact pressure (1N) or the oil used (mineral oil Oroil Therm7). However, it is necessary to duly justify why the 100Cr6 steel, a pressure of 1N or a mineral oil of a specific viscosity has been used instead of carrying out the study under other conditions. Please insert an explanation in the text.

2-. There is no clear justification for conducting the study on the patterns described in Figure 3. Why has it been decided to conduct the study on these and not others? What benefit effect is expected in each case? In a scientific study there must be a previous hypothesis about the expected behavior in each texture.

3-. While Figure 3 presents 7 different texturing patterns, sections 2.1.1 and 2.1.2 describe only 5 the textures Nautilus Fig3(a), Sunflower Fig 3(b), Diagonal Fig 3(e), Hydrostep Fig 3(f) and Fishbone Fig3(g), with no mention to the textures corresponding to Figures 3(c) and 3(d), which, as it turns out, correspond to the Grid Textures 33% and 44%. It is necessary to introduce a clarification in the text in point 2.

4-. Although the study evaluates the coefficient of friction of a reference non-textured surface, it is not mentioned in the text and should be explained in the text. Moreover, is not clear why the improvement obtained in each case is not quantified in a more specific way. The study shows variations from an untextured surface, but it is not clear whether this variation is relevant in terms of practical application and whether it justifies the extra cost of laser texturing.

5-. On the ordinate axis of Figure 7, unlike figures 12, 13 and 14, the letter µ is used instead of the abbreviation COF to refer to the coefficient of friction.

6-. On line 102 the word texture is misspelled.

Reviewer 4 Report

In the paper “Laser microtextured surfaces for friction reduction: does the pattern matter?” authors studied the frictional performance of the modification of the topography by laser texturing

Different textures have been obtained, and their performance have been studied in mixed and hydrodynamic lubrication regimes. Not only experimental results have been obtained, but also they have been corroborated by numerical simulations.

Methodology and materials are good explained, the conclusions are really interesting, and references are updated.

From my point of view, it is a very interesting and well elaborated work that should be published.

Only some comments about the paper:

-References 11 and 31 should be reviewed.

-Line 81: Typing mistake in units of parameter a.

Reviewer 5 Report

In the 'Materials and Methods' section, you need to enter a subsection on numerical tests, including a reference to 'Appendix A'.

It is also necessary to refer to the relationship of microdimples and roughness in the friction system, based on e.g.
Jing ‑ Hu Ji, Cai ‑ Wei Guan and Yong ‑ Hong Fu, Efect of Micro-Dimples on Hydrodynamic Lubrication of Textured Sinusoidal Roughness, Surfaces, Chin. J. Mech. Eng. (2018) 31:67, https://doi.org/10.1186/s10033-018-0272-z

To give a more complete picture of the role of microdimples, it is worth considering their influence on vibrations in the frictional system based on e.g.
U. Sudeep, N. Tandon, R. K. Pandey, Friction and vibration behaviors of lubricated laser textured point contacts under reciprocating rolling motion with highlights on the used laser parameters, Procedia Technology 14 (2014) 4 - 11, 2212-0173, 2nd International Conference on Innovations in Automation and Mechatronics Engineering, ICIAME 2014

Chen, C., Wang, X., Shen, Y., Li, Z. and Dong, J. (2019), "Experimental investigation for vibration reduction of surface-textured journal bearings", Industrial Lubrication and Tribology, Vol. 71 Well. 2, pp. 232-241. https://doi.org/10.1108/ILT-05-2018-0173

In the abstract, it is worth clearly formulating the text of the study.
In the caption for Figure 1, the signs 1, 2, 3, 4 should be explained - regardless of their indication in the text.
Explain the meaning of 'f' in formula (1). It is better to give, for example, some subscript to this notation, because of the similarity to that used in 'fs'.
In Figure 2, it would be a good idea to enter a reference length segment
Explain the meaning of 'θ' in formulas (4), (5) - a schematic drawing explaining the position of the system in relation to the analyzed microdimples would be useful
In formulas (6), (8), (9) to explain the meaning of 'n' - the text is also for laymen
Please, give the exact formula used for calculation of the geometric void ratio (VR), with a specific explanation of what is taken as the microdimples area and what is the entire surface area considered (is it a fragment of a spherical surface or its projection?)
Figures 5 and 6 - need clearer markings and numbers on the axes, especially on the 'z'
Figure 7 - Could do with adding an enlarged center section - attention to detail
In charts 7, 8, 12,13, 14 - remove the grid lines - they worsen the readability
Fig. 12 and 14 - explain the designation of the arrows directly in the drawing or in the figure caption
In the 'Results and Discussion' section, either separate subsections on experimental and numerical research, or specify in the description of the figures whether they refer to numerical or experimental results - otherwise it causes a bit of confusion.
It is better to avoid phrases like 'To this extent, we have started our study with symmetric textures, including the Nautilus, the Sunflower, the Fishbone and two uniform meshes with different geometric void ratios' in the 'Conclusion' section

Round 2

Reviewer 2 Report

Inappropriate font size in figure: 2,5,6 and 9

Reviewer 3 Report

The changes introduced clarify and correct the points pointed out in the review so the manuscript is now in a position to be accepted.